# FedMes: Speeding Up Federated Learning with Multiple Edge Servers

## Abstract

We consider federated learning with multiple wireless edge servers having their own local coverages. We focus on speeding up training in this increasingly practical setup. Our key idea is to utilize the devices located in the overlapping areas between the coverage of edge servers; in the model-downloading stage, the devices in the overlapping areas receive multiple models from different edge servers, take the average of the received models, and then update the model with their local data. These devices send their updated model to multiple edge servers by broadcasting, which acts as bridges for sharing the trained models between servers. Even when some edge servers are given biased datasets within their coverages, their training processes can be assisted by coverages of adjacent servers, through the devices in the overlapping regions. As a result, the proposed scheme does not require costly communications with the central cloud server (located at the higher tier of edge servers) for model synchronization, significantly reducing the overall training time compared to the conventional cloud-based federated learning systems. Extensive experimental results show remarkable performance gains of our scheme compared to existing methods.

## 1 Introduction

With the explosive growth in the numbers of smart phones, wearable devices and Internet of Things (IoT) sensors, a large portion of data generated nowadays is collected outside the cloud, especially at the distributed end-devices at the edge. Federated learning (McMahan et al., 2017; Konecny et al., 2016b;a; Bonawitz et al., 2019; Li et al., 2019a) is a recent paradigm for this setup, which enables training of a machine learning model in a distributed network while significantly resolving privacy concerns of the individual devices. However, training requires repeated downloading and uploading of the models between the parameter server (PS) and devices, presenting significant challenges in terms of 1) the communication bottleneck at the PS and 2) the nonIID (independent, identically distributed) data characteristic across devices (Zhao et al., 2018; Sattler et al., 2019; Li et al., 2019b; Reisizadeh et al., 2019; Jeong et al., 2018).

In federated learning, the PS can be located at the cloud or at the edge (e.g., small base stations). Most current studies on federated learning consider the former, with the assumption that millions of devices are within the coverage of the PS at the cloud; at every global round, the devices in the system should communicate with the PS (located at the cloud) for downloading and uploading the models. However, an inherent limitation of this cloud-based system is the long distance between the device and the cloud server, which causes significant propagation delay during model downloading/uploading stages in federated learning (Mao et al., 2017; Nguyen et al., 2019). Specifically, it is reported in (Mao et al., 2017) that the supportable latency (for inference) of cloud-based systems is larger than 100 milliseconds, while the edge-based systems have supportable latency of less than tens of milliseconds. This large delay between the cloud and the devices directly affects the training time of cloud-based federated learning systems. In order to support latency-sensitive applications (e.g., smart cars) or emergency events (e.g., disaster response by drones) by federated learning, utilization of edge-based system is absolutely necessary.

An issue, however, is that although the edge-based federated learning system can considerably reduce the latency between the PS and the devices, the coverage of an edge server is generally limited in practical systems (e.g., wireless cellular networks); there are insufficient number of devices within the coverage of an edge server for training a global model with enough accuracy. Accordingly, the

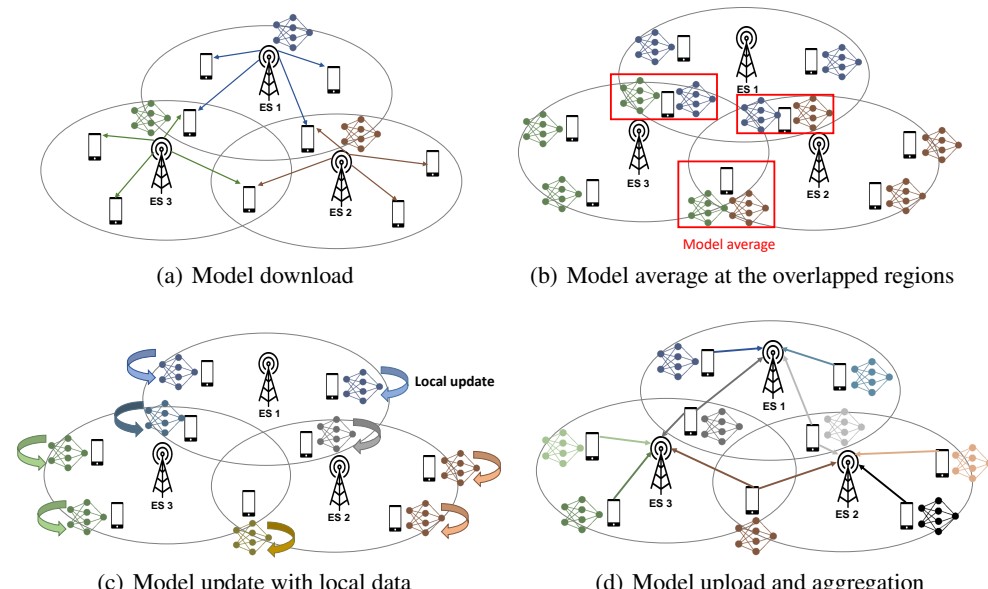

(a) Model download

(b) Model average at the overlapped regions

(c) Model update with local data

(d) Model upload and aggregation

Figure 1: FedMes: the proposed federated learning algorithm leveraging multiple edge servers (ESs). The devices in the overlapping areas can act as bridges for sharing the trained models between ESs. When a specific ES is given a biased dataset within its coverage, giving more weights to the devices located in the overlapping regions (in each aggregation step at the ESs) can further speed up training. Our design targets latency-sensitive applications where edge-based federated learning is essential, e.g., when a number of cars/drones should quickly adapt to the current situation by cooperation (federated learning) and make the right decision.

limited coverage of a single edge server could include biased datasets and thus could lead to a biased model after training. Thus in practice, performing federated learning with the devices in a single edge server would result in a significant performance degradation.

**Main contributions.** To overcome the above practical challenges, we propose FedMes, a novel federated learning algorithm highly tailored to the environment with multiple edge servers (ESs). Our idea here is to utilize the devices located in the overlapping areas between the coverage of ESs, which are typical in 5G and beyond systems with dense deployment of ESs. In the model-downloading stage, each ES sends the current model to the devices in its coverage area; in this process the devices in the overlapped region receive multiple models from different ESs. These devices in the overlapping area take the average of the received models, and then update the model based on their local data. Then each device sends its updated model to the corresponding ES or ESs, which is aggregated at each ES. A high-level description of FedMes is given in Fig. 1.

For example, suppose that device $k$ is located in the non-overlapped region of ES $i$ while device $l$ is in the overlapped region between ES $i$ and ES $j$. In conventional federated learning systems, device $l$ participates in the training process of only one of ES $i$ or ES $j$; on the other hand, in FedMes, device $l$ can act as a bridge for sharing the trained models between both ESs. To be specific, the updated model of device $k$ is averaged only at its associated ES $i$. In the next step, this averaged model is sent to the devices in its covered area, including device $l$. After the local model updates at the devices, device $l$ sends its updated model to both ES $i$ and ES $j$. From this point of view, even when some training samples are only in the coverage of a specific ES, these data can still assist the training process of other servers. Hence, the proposed scheme does not require costly communications with the central cloud server (located at the higher tier of ESs) for model synchronization, significantly reducing the overall training time compared to cloud-based federated learning systems. Comparing with the scheme which does not consider the overlapping areas, FedMes can provide a significant performance gain, especially when the data distributions across coverages of different servers are nonIID, e.g., when a specific server has a biased dataset within its covered area. Especially in this nonIID setup, giving more weights to the devices located in the overlapping areas (in each aggregation step at the ESs) can further speed up training.

From the service provider point of view, FedMes does not require any backhaul traffic between the ESs and the cloud server, significantly reducing the communication resources required for federated

learning. Moreover, since the devices in the overlapping areas send their results to *multiples ESs*, our scheme can reduce the number of devices participating at each global round while achieving the desired performance.

Extensive experimental results on various datasets show that FedMes provides remarkable performance gain compared to 1) the scheme that requires communications with the central cloud server for model synchronization (i.e., cloud-based federated learning) and 2) the scheme that does not take the overlapping areas between servers into account.

**Related works.** Thanks to the recent advent of edge computing, there has been an increased interest in edge-facilitated federated learning systems (Tran et al., 2019; Wang et al., 2019; Lim et al., 2020; Abad et al., 2020; Liu et al., 2019). The authors of (Wang et al., 2019) focused on optimizing federated learning framework with a given resource budget in wireless edge networks. The authors of (Tran et al., 2019) considered resource allocation to minimize energy consumption at the devices in wireless networks. However, a single-server setup is considered in (Tran et al., 2019) and (Wang et al., 2019), which is totally different from our work leveraging multiple edge servers. Only a few prior works on federated learning (Abad et al., 2020; Liu et al., 2019) considered a setup with multiple edge servers. However, the schemes of (Abad et al., 2020; Liu et al., 2019) still require costly communication with the central cloud server for model synchronization, which could significantly slow down the overall training process. If the communication period with the cloud is small, frequent model synchronization between edge servers is possible but incurs a large communication time delay. Infrequent model synchronization can lead to a bad performance especially with nonIID data across the servers. FedMes overcomes these challenges by enabling fast federated learning without any help with the cloud server, i.e., only using the edge servers. It is shown later in Section 4 that our scheme can outperform the hierarchical scheme of (Liu et al., 2019) which requires costly communications with the central cloud server.

## 2 PROBLEM SETUP

**Federated learning.** Let $K$ be the number of devices in the system. Let $n_k$ be the number of data samples in device $k$, with $n = \sum_{k=1}^{K} n_k$ being the total number of training samples. We also denote the $i$-th sample in device $k$ as $x_{k,i}$, for $i \in \{1, 2, ..., n_k\}$. Our goal is to solve the following optimization problem

$$\min_{\mathbf{w}} F(\mathbf{w}) = \min_{\mathbf{w}} \sum_{k=1}^{K} \frac{n_k}{n} F_k(\mathbf{w}), \tag{1}$$

where $F_k(\mathbf{w})$ is the local loss function of data samples in device $k$, written as $F_k(\mathbf{w}) = \frac{1}{n_k} \sum_{i=1}^{n_k} \ell(x_{k,i}; \mathbf{w})$. Now we briefly describe the conventional *cloud-based federated averaging (FedAvg) algorithm* in (McMahan et al., 2017), a typical way to solve this problem. At step $t$, each device downloads the current model $\mathbf{w}(t)$ from the PS, which is generally located at the cloud covering all the devices in the system. Then each device (say device $k$) sets $\mathbf{w}_k(t) = \mathbf{w}(t)$ and performs $E$ local updates according to

$$\mathbf{w}_k(t+i+1) = \mathbf{w}_k(t+i) - \eta_{t+i} \nabla F_k(\mathbf{w}_k(t+i), \xi_k(t+i)), \ i = 0, 1, ..., E-1, \tag{2}$$

where $\eta_t$ is the learning rate at step $t$ and $\xi_k(t)$ is a set of randomly selected data samples from device $k$ at step $t$. Now each device sends the updated model to the PS, and the PS aggregates the model as $\mathbf{w}(t+E) = \sum_{k=1}^{K} \frac{n_k}{n} \mathbf{w}_k(t+E)$. However, full device participation at each aggregation step is impossible in practice and the PS often selects a set $S_{t+E} \subset \{1, 2, ..., K\}$, containing the devices that transmit the results to the PS. Then, we have

$$\mathbf{w}(t+E) = \sum_{k \in S_{t+E}} \frac{n_k}{n} \mathbf{w}_k(t+E). \tag{3}$$

This overall process is repeated until the model achieves the desired accuracy or some stopping condition is met. According to the algorithm, the model of the $k$-th device at step $t+1$ is written as

$$\mathbf{w}_k(t+1) = \begin{cases} \mathbf{w}_k(t) - \eta_t \nabla F_k(\mathbf{w}_k(t), \xi_k(t)), & \text{if } E \nmid t+1 \\ \sum_{q \in S_{t+1}} \frac{n_q}{n} [\mathbf{w}_q(t) - \eta_t \nabla F_q(\mathbf{w}_q(t), \xi_q(t))], & \text{otherwise.} \end{cases} \tag{4}$$

**Problem formulation.** In contrast with the conventional cloud-based federated learning systems having a central cloud server covering the whole devices, in this paper we consider $L$ ESs each

covering its own local area. We call this local coverage of each edge server a *cell*. Especially with dense deployment of ESs in 5G and beyond networks, there generally exist more than one ES within the range of a specific user that can be reliably communicate with. We call this region in which the device can reliably communicate with multiple ESs *overlapping cell area*. Let $C_i$ be the set of indices for users located in cell $i \in \{1, 2, ..., L\}$. Now define $U_i$ as the set of user indices for the non-overlapped region of cell $i$, which is the subset of $C_i$. We also define $V_{i,j}$ as the set of user indices for the overlapping area between cell $i$ and cell $j$ ($i \neq j$ and $V_{i,j} = V_{j,i}$), which is also the subset of $C_i$. Here, the devices in $V_{i,j}$ can communicate with both ES $i$ and ES $j$ during model download or upload. While we can similarly define overlapped regions with more than two ESs, we consider the case in which the coverage of at most two ESs overlapped for clarity of presentation. Then the coverage of cell $i$, i.e., $C_i$ can be written as

$$C_i = U_i \cup \left( \bigcup_{j \in [L] \setminus \{i\}} V_{i,j} \right) \tag{5}$$

for all $i \in \{1, 2, ..., L\}$. The overall coverage of the system can be written as $C = \{1, 2, ..., K\} = \cup_{i=1}^{L} U_i \cup (\cup_{i=1}^{L} \cup_{j=i+1}^{L} V_{i,j})$. Note that each ES can communicate with the devices in its covered area only. Our goal is to solve the problem in (1) in this setup, without sharing the learning models at the higher tier of ESs (i.e., the cloud server) during training.

## 3 PROPOSED FEDMES ALGORITHM

Now we describe FedMes, the proposed algorithm tailored to the above setup with multiple edge servers. At the beginning of step $t$, each device in cell $i$ downloads the current model $\mathbf{w}^{(i)}(t)$ from ES $i$ and lets $\mathbf{w}_k(t) = \mathbf{w}^{(i)}(t)$ if located in the non-overlapped region, i.e., $k \in U_i$. Device $k \in V_{i,j}$ additionally downloads $\mathbf{w}^{(j)}(t)$ from ES $j$ and combines the two received models according to

$$\mathbf{w}_k(t) = \frac{1}{\sum_{k \in S_t^{(i)}} n_k + \sum_{k \in S_t^{(j)}} n_k} \left( \sum_{k \in S_t^{(i)}} n_k \mathbf{w}^{(i)}(t) + \sum_{k \in S_t^{(j)}} n_k \mathbf{w}^{(j)}(t) \right), \tag{6}$$

where $S_t^{(i)}$ and $S_t^{(j)}$ are the sets of devices that sent their results to ES $i$ and ES $j$, respectively, at the previous aggregation step. Here, a larger weight is given to the aggregated model of the ES that utilized more training samples in the previous aggregation step. If the number of training samples utilized at both ES $i$ and ES $j$ are the same, i.e., $\sum_{k \in S_t^{(i)}} n_k = \sum_{k \in S_t^{(j)}} n_k$, then we can rewrite (6) as

$$\mathbf{w}_k(t) = \frac{1}{2}(\mathbf{w}^{(i)}(t) + \mathbf{w}^{(j)}(t)). \tag{7}$$

We take this assumption for the rest of the paper for ease of presentation. Now every device (say device $k$) updates the model with its own local data according to equation (2) for $E$ steps to obtain $\mathbf{w}_k(t + E)$. Then, device $k$ sends its updated model $\mathbf{w}_k(t + E)$ to the corresponding ES(s). In particular, the devices in the overlapped region $V_{i,j}$ send their results to both ES $i$ and ES $j$ by *broadcasting*. Here, we note that this does not require additional communication time since only one round of communication is required at each device by the broadcast nature of wireless networks.

Now every ES $i$ collects the locally updated models from the devices in its covered area $C_i$, to obtain the aggregated result. In this aggregation process, each ES takes the weighted average of the received models, where the weight depends on the location of device $k$ (i.e., whether device $k$ is located in the overlapped region or not). To be specific, let $S_{t+E}^{(i)}$ be the set of indices for devices that transmit the results to ES $i$. Then, ES $i$ obtains the aggregated result $\mathbf{w}^{(i)}(t + E)$ by taking the weighted average of the received models as

$$\mathbf{w}^{(i)}(t + E) = \sum_{k \in S_{t+E}^{(i)}} \gamma_k \mathbf{w}_k(t + E), \tag{8}$$

where $\gamma_k$ is the normalized parameter that gives weight depending on device $k$'s location and the number of data samples in it:

$$\gamma_k = \begin{cases} \alpha_u n_k, & k \in U_i \\ \alpha_v n_k, & k \in V_{i,j} \text{ for all } j \end{cases} \quad \text{and} \quad \sum_{k \in S_{t+E}^{(i)}} \gamma_k = 1. \tag{9}$$

Here, $\alpha_u$ and $\alpha_v$ are constant values and we generally consider the case with $\alpha_u \leq \alpha_v$ to give larger weights to the devices in the overlapped regions that can act as bridges for sharing the trained models

---

**Algorithm 1** Federated Learning with Multiple Edge Servers (FedMes)

---

**Input:** Initialized model $\mathbf{w}(0)$, **Output:** Final global model $\mathbf{w}^f$
Set $\mathbf{w}_k(0) = \mathbf{w}(0)$ for all devices $k = 1, 2, ..., K$

1: **for** each step $t = 0, 1, ..., T - 1$ **do**
2:    **if** $E \nmid t + 1$ **then**
3:      **for** each device $k = 1, 2, ..., K$ in parallel **do**
4:        $\mathbf{w}_k(t + 1) \leftarrow \mathbf{w}_k(t) - \eta_t \nabla F_k(\mathbf{w}_k(t), \xi_k(t))$    // Local update
5:      **end for**
6:    **else**
7:      Each edge server $i$ randomly selects $S_{t+1}^{(i)}$ to receive the updated models from the devices
8:      **for** each device $k = 1, 2, ..., K$ in parallel **do**
9:        **if** $k \in U_i$ for some $i \in \{1, 2, ..., L\}$ **then**
10:          $\mathbf{w}_k(t + 1) \leftarrow \sum_{k \in S_{t+1}^{(i)}} \gamma_k[\mathbf{w}_k(t) - \eta_t \nabla F_k(\mathbf{w}_k(t), \xi_k(t))]$    // Edge aggregation
11:        **end if**
12:        **if** $k \in V_{i,j}$ for $i, j \in \{1, 2, ..., L\}$ and $i \neq j$ **then**
13:          $\mathbf{w}_k(t + 1) \leftarrow \frac{1}{2}\Big( \sum_{k \in S_{t+1}^{(i)}} \gamma_k[\mathbf{w}_k(t) - \eta_t \nabla F_k(\mathbf{w}_k(t), \xi_k(t))]$    // Average of

              $+ \sum_{k \in S_{t+1}^{(j)}} \gamma_k[\mathbf{w}_k(t) - \eta_t \nabla F_k(\mathbf{w}_k(t), \xi_k(t))]\Big)$    two edge aggregations
14:        **end if**
15:      **end for**
16:    **end if**
17: **end for**
18: $\mathbf{w}^f \leftarrow \frac{1}{L} \sum_{i=1}^{L} \sum_{k \in S_T^{(i)}} \gamma_k \mathbf{w}_k(T)$   // Averaged model of $L$ edge servers

---

between ESs. Note that by setting $\alpha_u = \alpha_v$, the above aggregation rule in (8) at each ES reduces to the conventional FedAvg in (McMahan et al., 2017). This overall process at the edge servers and the devices is repeated until some stopping condition is met, say at step $T$. When the overall training is finished, the models are averaged over all ESs to obtain the final global model $\mathbf{w}^f$ as follows:

$$\mathbf{w}^f = \frac{1}{L} \sum_{i=1}^{L} \mathbf{w}^{(i)}(T). \tag{10}$$

The details of FedMes are summarized in Algorithm 1. Compared to the conventional cloud-based federated learning systems which require communication with the cloud server (located at the higher tier of ESs) for model synchronization, in FedMes, only the communications between the devices and ESs are required. Fig. 1 shows an example with $L = 3$, $|U_1| = |U_2| = |U_3| = 2$ and $|V_{1,2}| = |V_{2,3}| = |V_{3,1}| = 1$. As can be seen from Figs. 1(b) and 1(c), the unique characteristic of FedMes is that the starting points of models could be different even for the devices in the same cell, due to model average in the overlapped regions. This is totally different from typical federated learning algorithms in which the devices have the same starting points at the beginning of each global round. From the overall process, it can be also seen that even when some training samples are only in the non-overlapped region of cell $i$, i.e., $U_i$, these data can still assist the training process of other cells. We show later in Section 4 that our scheme with multiple starting points performs well by sufficiently reflecting the data samples that are not in the coverage of a specific ES. By utilizing the weight parameters $\gamma_k$ in (9), the model of device $k \in C_i$ at step $t + 1$ can be written as follows:

$$\mathbf{w}_k(t + 1) = \begin{cases} \mathbf{w}_k(t) - \eta_t \nabla F_k(\mathbf{w}_k(t), \xi_k(t)), & \text{if } E \nmid t + 1 \\ \displaystyle\sum_{k \in S_{t+1}^{(i)}} \gamma_k[\mathbf{w}_k(t) - \eta_t \nabla F_k(\mathbf{w}_k(t), \xi_k(t))], & \text{if } E \mid t + 1 \text{ and } k \in U_i \\ \frac{1}{2}\Big( \displaystyle\sum_{k \in S_{t+1}^{(i)}} \gamma_k[\mathbf{w}_k(t) - \eta_t \nabla F_k(\mathbf{w}_k(t), \xi_k(t))] \\ \quad + \displaystyle\sum_{k \in S_{t+1}^{(j)}} \gamma_k[\mathbf{w}_k(t) - \eta_t \nabla F_k(\mathbf{w}_k(t), \xi_k(t))]\Big), & \text{if } E \mid t + 1 \text{ and } k \in V_{i,j} \end{cases} \tag{11}$$

for $i \in \{1, 2, ..., L\}$ and $j \in \{1, 2, ..., L\} \setminus \{i\}$.

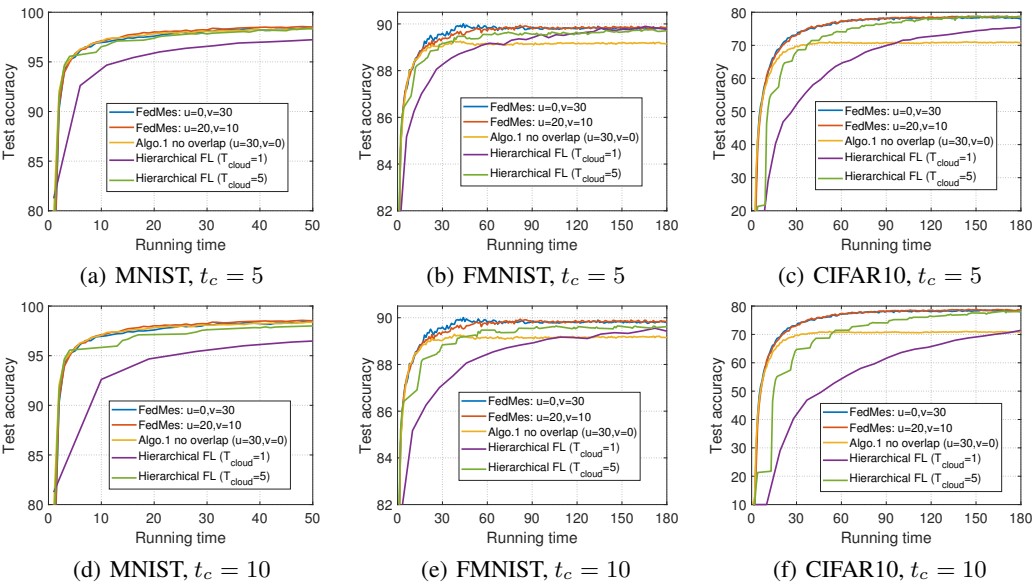

Figure 2: Test accuracy versus training time in a (device IID, cell IID) setup.

**Remark 1.** *Note that our FedMes can be easily generalized to cell topologies with multiple overlapping ESs (more than two), by simply adding equations in (11) for the regions with 3, 4 overlapping ESs or more. We also note that (11) holds regardless of the device's location, which means that (11) is still valid even with moving devices (i.e., mobiles).*

## 4 EXPERIMENTS

In this section, we provide experimental results for our algorithm using the MNIST (LeCun et al., 1998), FMNIST (Xiao et al., 2017) and CIFAR10 (Krizhevsky et al., 2009) datasets. We split the MNIST and FMNIST datasets into 60,000 training samples and 10,000 test samples, and split the CIFAR10 dataset into 50,000 training samples and 10,000 test samples. For MNIST and FMNIST, we utilized the convolutional neural network (CNN) with 2 convolutional layers and 2 fully connected layers, and 2 convolutional layers and 1 fully connected layer, respectively. For CIFAR10, we utilized VGG-11 for training.

**Comparison schemes.** We compare our FedMes with the following schemes. First, we consider the *hierarchical FL* scheme (Liu et al., 2019), which utilizes $L$ edge servers (with no overlapping regions) and a single cloud server. After $E$ local updates at each device, each edge server aggregates the models from the devices in its own covered area and then distributes across the corresponding devices. After $T_{cloud}$ aggregations at each edge server, the cloud server aggregates the models from the edge servers and distributes across the devices. Here, the case with $T_{cloud} = 1$ can be viewed as the cloud-based systems since communication with the cloud occurs with every $E$ local updates at the devices. Secondly, we consider *Algorithm 1 with no overlap*, where the proposed Algorithm 1 is applied in a setup with no overlapping cell regions. Each edge server independently performs training without any communication with the cloud, i.e., $T_{cloud} = \infty$. When training is finished, the model parameters of different cells are averaged to obtain the final global model.

**Experimental setup.** We consider a setup with $L = 3$ edge servers and $K = 90$ devices. We also consider a symmetric cell geometry with $|U_1| = |U_2| = |U_3| = u$ and $|V_{1,2}| = |V_{2,3}| = |V_{3,1}| = v$, where $u + v = K/3 = 30$ holds. In the aggregation step at the edge servers, we let each edge server to collect the results of randomly selected $m = 20$ devices in its covered area. Considering $L = 3$ edge servers, a total of 60 devices participate in each aggregation step when there are no overlapping regions ($v = 0$). With $v > 0$, each edge server has access to $u + 2v$ devices. For a fair comparison, we let each edge server to collect the results of $m = 20$ devices, where $\frac{um}{u+2v}$ of them are taken from the non-overlapped region $U_i$, and $\frac{2vm}{u+2v}$ of them are selected from the two overlapping areas, $\frac{vm}{u+2v}$ of each. Assuming that the adjacent edge servers select the same $\frac{vm}{u+2v}$ devices from the overlapping area by cooperation, a total of $\frac{3(u+v)m}{u+2v} = \frac{60(u+v)}{u+2v}$ devices participates in each aggregation step. Hence, the number of devices participating at each global round is reduced compared to the scheme with no

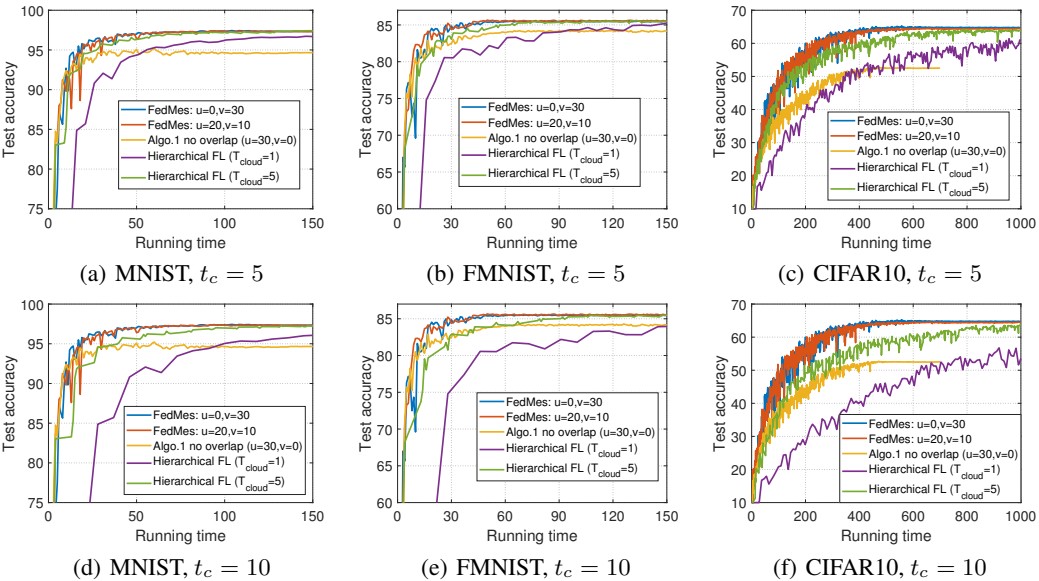

Figure 3: Test accuracy versus training time in a (device nonIID, cell IID) setup.

overlap. We let $t_c$ be the one-round delay between the device and the cloud, where the one-round delay between the device and the edge server is normalized to 1. This one-round delay include both the communication time for one-round trip between the server and the device, and the computation time at each device. Mini-batch stochastic gradient descent with an initial learning rate of 0.01 is utilized. For all experiments, we set the number of local epochs at each device to 5, and the local mini-batch size to 10.

We consider three data allocation scenarios to test nonIID situations: The first is the (device IID, cell IID) case, where the datasets across $K$ devices and across $L$ cells are IID. Secondly, we consider (device nonIID, cell IID) case where each device randomly selects two classes for data assignment. Hence, the datasets across devices are nonIID but across cells are IID. The last is the (device nonIID, cell nonIID) where the devices in each cell are given with two fixed classes. Hence, the datasets across devices and cells are both nonIID. For the first two scenarios, we only consider the case with $\alpha_u = \alpha_v$ since our scheme with $\alpha_u = \alpha_v$ already shows the optimal performance in these scenarios. For the last nonIID scenario, we consider both cases with $\alpha_u = \alpha_v$ and $\alpha_u < \alpha_v$. In addition, we ignored the batch normalization layers of the VGG model during training and testing for the last two nonIID scenarios.

**Experiments in a (device IID, cell IID) setup.** In Fig. 2, we plot the test accuracy versus running time in a (device IID, cell IID) setup. We let $\alpha_u = \alpha_v$ for FedMes. We have the following important observations in this setup. First, the proposed schemes with $v > 0$ have very similar performances. It can be seen that FedMes with $u = 20$, $v = 10$ has almost the same performance as the scheme with all devices in the overlapping areas ($u = 0$, $v = 30$). Comparing FedMes with *Algorithm 1 with no overlap* ($u = 30$, $v = 0$), we have the following results. For the MNIST dataset, Algorithm 1 with no overlap performs almost the same with the proposed schemes, since the dataset is simple enough to achieve the ideal performance by only utilizing the data covered by a specific cell. With the FMNIST dataset, a slight gap between FedMes and Algorithm 1 with no overlap is observed. For a more complicated dataset CIFAR10, we have a larger gap between them, around test accuracy of $8\%$. Since the proposed FedMes and Algorithm 1 with no overlap do not require any communication with the central cloud server during training, the performances are not affected by the parameter $t_c$.

Now we observe the *hierarchical FL* (Liu et al., 2019) with $T_{cloud} = 1$, where the edge servers communicate with the cloud for model synchronization in every aggregation step at the edges. For all three datasets, our observation is that FedMes can eventually achieve the best-test accuracy of the cloud-based scheme, which is the ideal performance. However, the proposed schemes do not require costly communication with the central cloud server, significantly reducing the running time to achieve the ideal test accuracy compared to the cloud-based systems. The hierarchical FL with $T_{cloud} > 1$ can reduce the period of communication with the cloud, while achieving the ideal performance. Hence,

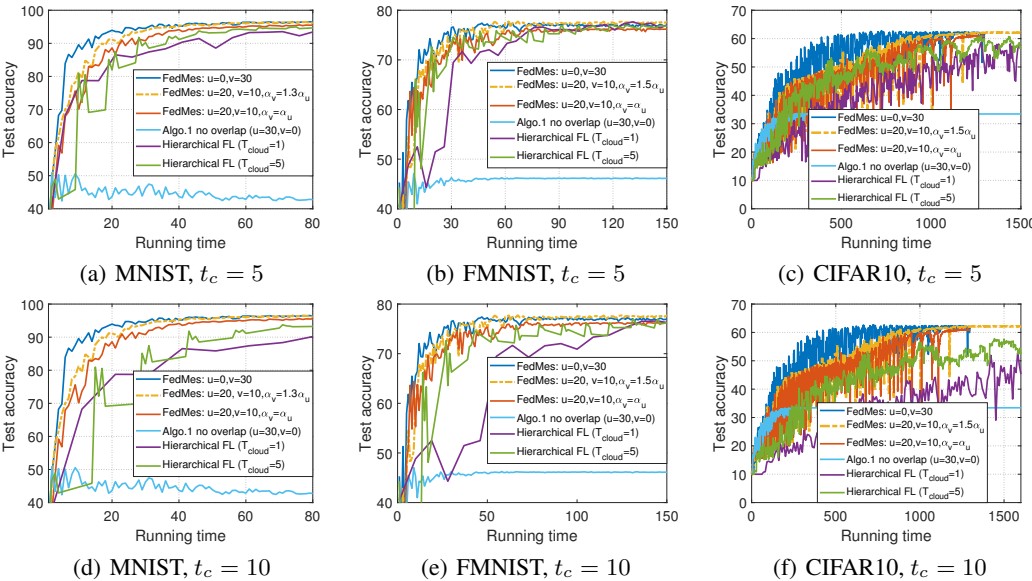

Figure 4: Test accuracy versus training time in a (device nonIID, cell nonIID) setup.

with a properly chosen $T_{cloud}$, it can be seen that hierarchical FL with $T_{cloud} > 1$ performs better than the cloud-based scheme with $T_{cloud} = 1$. However, the hierarchical FL scheme still suffers from communication delay with the cloud, especially when $t_c$ is large. Here, we note that a very large $T_{cloud}$ can reduce the communication time with the cloud in the hierarchical scheme, but as $T_{cloud}$ grows, it gets closer to the comparison scheme *Algorithm 1 with no overlap*. Therefore, significant degradation of the accuracy with complicated datasets is expected when a large $T_{cloud}$ is chosen to reduce the communication time with the cloud. The overall results show remarkable performance gains of FedMes compared to others, especially with on CIFAR10.

**Experiments in a (device nonIID, cell IID) setup.** Fig. 3 shows the result in a (device nonIID, cell IID) setup. We set $\alpha_u = \alpha_v$ for our schemes. Since the datasets across the devices are nonIID, the performance of each scheme is degraded compared to the case in Fig. 2. We also observe a larger gap between the proposed schemes and Algorithm 1 with no overlap. The trend is consistent with the (device IID, cell IID) case, confirming the advantage of FedMes utilizing the devices in the overlapping cell areas.

**Experiments in a (device nonIID, cell nonIID) setup.** Finally, the case with nonIID datasets across both devices and cells is illustrated in Fig. 4. In this nonIID setup, we consider both cases with $\alpha_u = \alpha_v$ and $\alpha_u < \alpha_v$ for FedMes. An interesting observation here is that the test accuracy of hierarchical FL with $T_{cloud} = 5$ decreases before each aggregation step at the cloud. This is because each edge server is given a biased dataset within its coverage due to the cell nonIIDness, which can degrade the performance before model synchronization at the central cloud. Since *Algorithm 1 with no overlap* does not allow any synchronization at the cloud during training, it has significantly low test accuracy. In FedMes, the devices in the overlapping cell areas enable to share the trained models between edge servers. Therefore, even when some edge servers have biased datasets within their coverage, the model of each edge server can be assisted by various classes that are not within its coverage. Especially in this scenario with nonIID data across cells, it is observed that giving more weights to the devices in the overlapping cell areas (i.e., $\alpha_u < \alpha_v$) can further speed up training than the scheme that gives the same weights to all devices ($\alpha_u = \alpha_v$). The overall results show that FedMes is still powerful even in this nonIID setup, compared to the schemes that require costly communication with the central cloud server for model synchronization and the scheme that does not take the overlapping areas between edge servers into account.

**Comparison with varying $t_c$.** We note that whatever computation/communication delay model we use, one can define the ratio of (one-round delay between the devices and the cloud) to (one-round delay between the device and the edge) as $t_c$, based on their system parameters. Now the question is, given a $t_c$ value of the current system, which scheme performs the best and how much performance gain do we have? Fig. 5 provides the result on test accuracy at a specific time as a function of $t_c$.

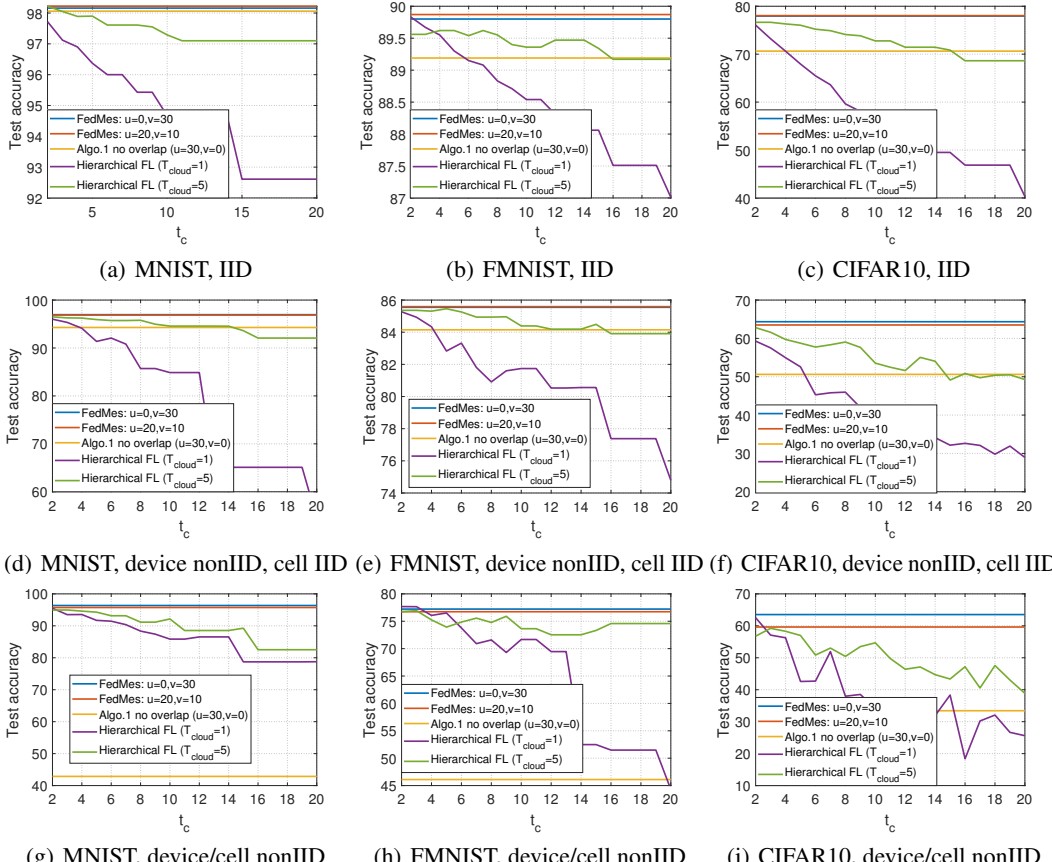

(a) MNIST, IID       (b) FMNIST, IID       (c) CIFAR10, IID

(d) MNIST, device nonIID, cell IID (e) FMNIST, device nonIID, cell IID (f) CIFAR10, device nonIID, cell IID

(g) MNIST, device/cell nonIID     (h) FMNIST, device/cell nonIID     (i) CIFAR10, device/cell nonIID

Figure 5: Test accuracy versus $t_c$. The test accuracies are computed at a specific time slot. 1) IID: time 30, 80, 80 for MNIST, FMNIST, CIFAR10. 2) (device nonIID, cell IID): time 40, 80, 400 for MNIST, FMNIST, CIFAR10. 3) device/cell nonIID: time 60, 80, 1000 for MNIST, FMNIST, CIFAR10.

It can be seen that FedMes performs the best even with a small $t_c$ value. Especially in practical regimes where the supportable latency of cloud-based systems are much larger than that of edge-based systems, FedMes can provide significant advantage compared to other baselines.

**Comparison with varying $v$.** The portion of devices in the overlapping area is another key metric that determines the performance of FedMes. In Supplementary Materials, we compare the performance of FedMes with different $v$ values. A quick conclusion is that although there are some slight performance degradation, FedMes performs well even with a small portion of devices in the overlapping regions (i.e., small $v$). Especially in 5G and beyond systems where a large number of devices are in the overlapping regions, FedMes can provide significant advantage compared to other schemes.

## 5 CONCLUSION

In this paper, we proposed FedMes, a federated learning algorithm leveraging multiple edge servers, to speed up training without sharing the models at the higher tier of edge servers. Our key idea was to utilize the devices in the overlapping cell areas, which act as bridges for sharing the models between servers. Extensive experiments verified the advantage of FedMes compared to existing methods on various datasets with different data distribution setups. Our solution enables to support real-time applications by speeding up federated learning in real-world wireless networks 1) having large communication time delay between the devices and the cloud server and 2) having biased datasets within the coverage of each edge server.

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

## A    TEST ACCURACY VERSUS GLOBAL ROUND

Fig. A.1 shows the accuracy-versus-round plots for each dataset and data distribution setup. It can be seen that FedMes with both ($u = 0$, $v = 30$) and ($u = 20$, $v = 10$) achieve the same final accuracy with the cloud-based system (purple line), except for the case with Fig. A.1(i). For CIFAR10 in a (device nonIID, cell nonIID) setup, FedMes with ($u = 0$, $v = 30$) achieves the same final accuracy with the purple line but the case with ($u = 20$, $v = 10$) does not. This indicates that a larger number of devices should be in the overlapping areas to act as bridges, especially when the data distributions across the cells are nonIID and the dataset is relatively complex.

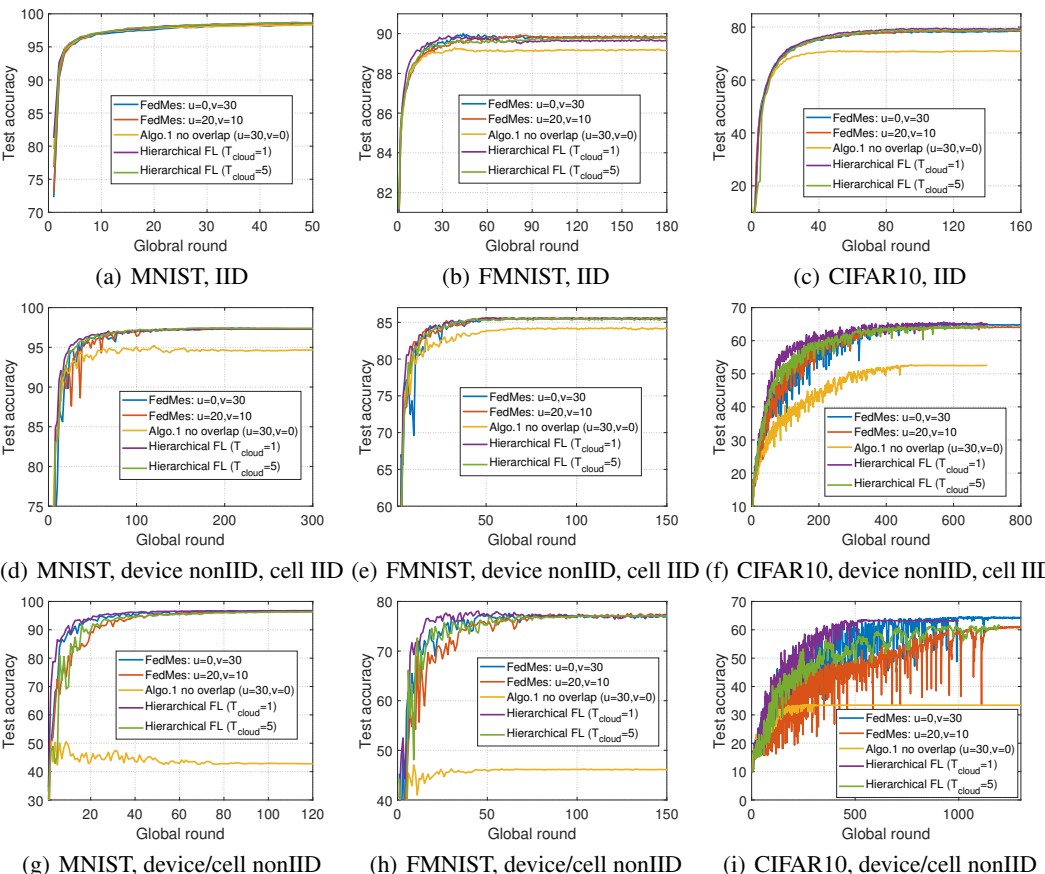

Figure A.1: Test accuracy versus global round.

## B    COMPARISON WITH VARYING $v$

In Fig. B.1, we compare the performance of FedMes with the following 4 cases: ($u = 0$, $v = 30$), ($u = 20$, $v = 10$), ($u = 25$, $v = 5$), ($u = 30$, $v = 0$). Although there are some slight performance degradation, FedMes performs well even with a small portion of devices in the overlapping regions.

## C    EFFECT OF VARYING $\alpha_u$, $\alpha_v$

In the main manuscript, we showed that giving more weights to the devices in the overlapping regions (during aggregation step at the edge servers) can further speed up FedMes, especially in a (device nonIID, cell nonIID) setup. In this section, we provide additional experiments to gain insights on the effect of varying $\alpha_u$ and $\alpha_v$. Consider $L = 3$ edge servers (ESs) and let $u$ be the number of devices in the non-overlapped region of each cell and $v$ be the number of devices in the overlapped region between any two cells.

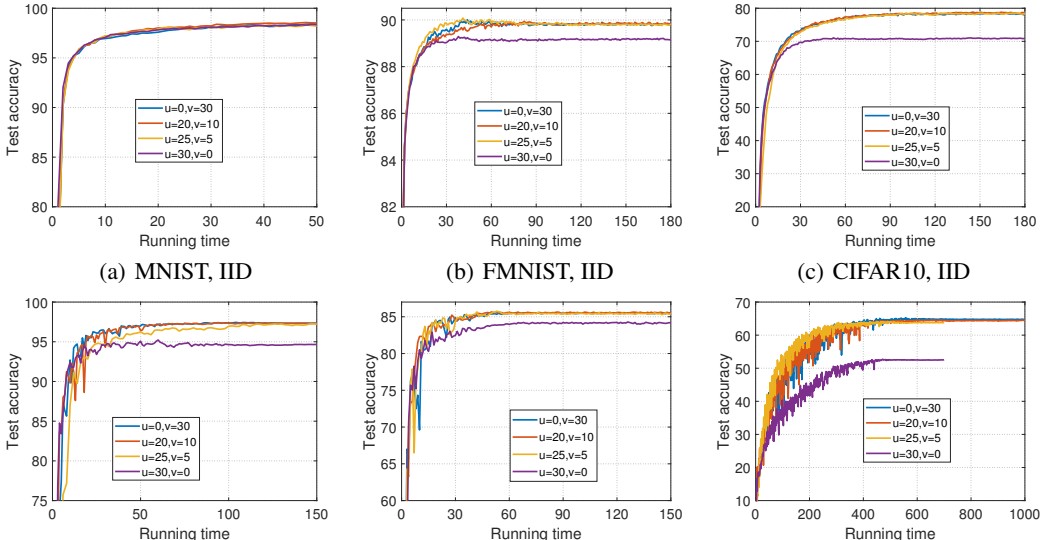

(a) MNIST, IID      (b) FMNIST, IID      (c) CIFAR10, IID

(d) MNIST, device nonIID, cell IID (e) FMNIST, device nonIID, cell IID (f) CIFAR10, device nonIID, cell IID

Figure B.1: Performance of FedMes with varying $v$.

Fig. C.1 shows the performance of FedMes with $u = 20$, $v = 10$ in a (device nonIID, cell nonIID) setup. The overall settings are exactly the same as in the main manuscript. We have the following interesting observations from Fig. C.1. First, if $\alpha_v$ is too small, i.e., if we give too small weights to the devices in the overlapped regions, the effect of devices in the overlapping areas are neglected; the aggregated models of the ESs cannot be shared through the devices in the overlapping areas, degrading the performance of learning. If $\alpha_v$ is too large, i.e., if we give large enough weights to the devices in the overlapped regions, the effect of devices in the non-overlapped regions are neglected which also significantly degrades the performance of the trained model. With an appropriate choice of $\alpha_u$ and $\alpha_v$, we can speed up training compared to the simple approach that gives the same weights to all devices in the system ($\alpha_u = \alpha_v$). In particular, for MNIST, $\alpha_v = 1.3\alpha_u$ gives the best performance, while $\alpha_v = 1.5\alpha_u$ does it for FMNIST and CIFAR10.

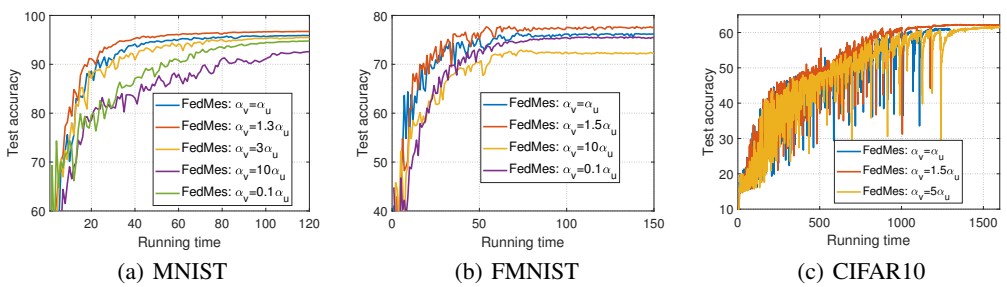

(a) MNIST      (b) FMNIST      (c) CIFAR10

Figure C.1: Effect of varying $\alpha_u$, $\alpha_v$ in a (device nonIID, cell nonIID) setup.

Fig. C.2 shows the result of FedMes with $u = 20$, $u = 10$ in a (device IID, cell IID) setup. For MNIST, the test accuracy is not affected by $\alpha_u$, $\alpha_v$; even though some devices may be neglected due to large/small $\alpha_v$, we can achieve ideal performance since the dataset is simple enough and data distributions across the devices are IID. For FMNIST, we see slight gaps among the schemes with different $\alpha_u$, $\alpha_v$ values. On CIFAR10, accuracies jump quite a bit as $\alpha_u$ and $\alpha_v$ differ significantly. Compared to the result in a nonIID setup in Fig. C.1, giving more weights to the devices in the overlapped region does not provide additional performance gain in this IID setup.

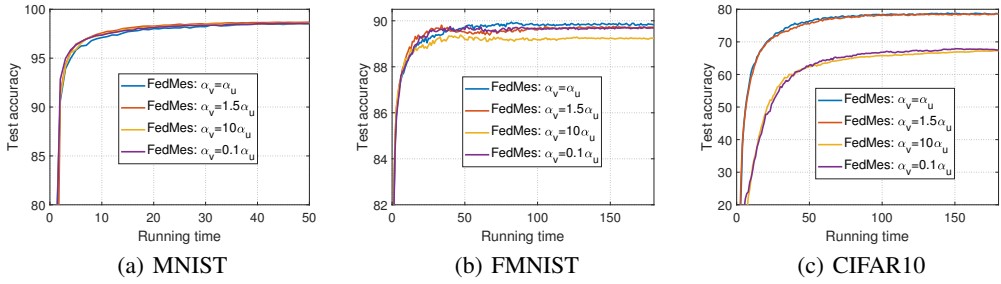

Figure C.2: Effect of varying $\alpha_u$, $\alpha_v$ in a (device IID, cell IID) setup.

# D    EXPERIMENTS WITH MULTIPLE OVERLAPPING ESS

In the experiment section of the main manuscript, we considered a setup where each device is located in the overlapping region of no more than two ESs. Here, we assume that the devices can be in regions with more than two overlapping ESs.

Again, we focus on $L = 3$ ESs and define $u$, $v$ as above. We also introduce $w$, the number of devices in the overlapped region among all $L = 3$ cells (see Fig. D.1). The number of devices in the system can be written as $K = 3(u + v) + w$. Fig. D.2 shows the performance of FedMes depending on $u$, $v$ and $w$, in a (device nonIID, cell nonIID) setup. Here, we let each ES to give the same weights to all devices regardless of the location. The overall results show that having more devices in the overlapped regions can speed up training. On CIFAR10, when $u > 0$, the case with $w > 0$ can achieve a target accuracy much faster.

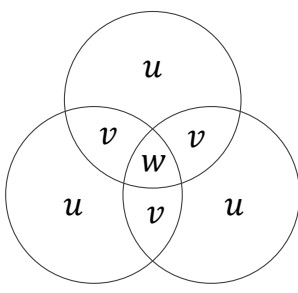

Figure D.1: Number of devices located in each region with $L = 3$ cells.

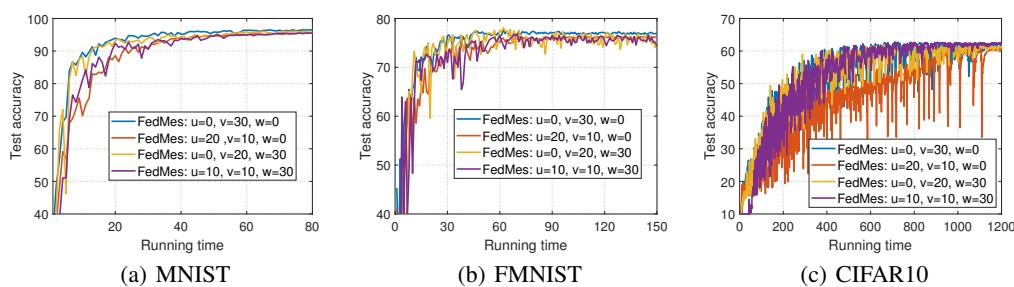

Figure D.2: Experiments with multiple overlapping ESs in a (device nonIID, cell nonIID) setup.

# E  EXPERIMENTS IN AN ASYMMETRIC CELL TOPOLOGY

In Fig. E.1, we provide additional experimental results in an asymmetric cell topology. The number of edge servers are $L = 3$, and we set the number of devices in each region as in Fig. E.2. The results are consistent with the plots in the main manuscript, confirming the advantage of FedMes in practical asymmetric cell topology.

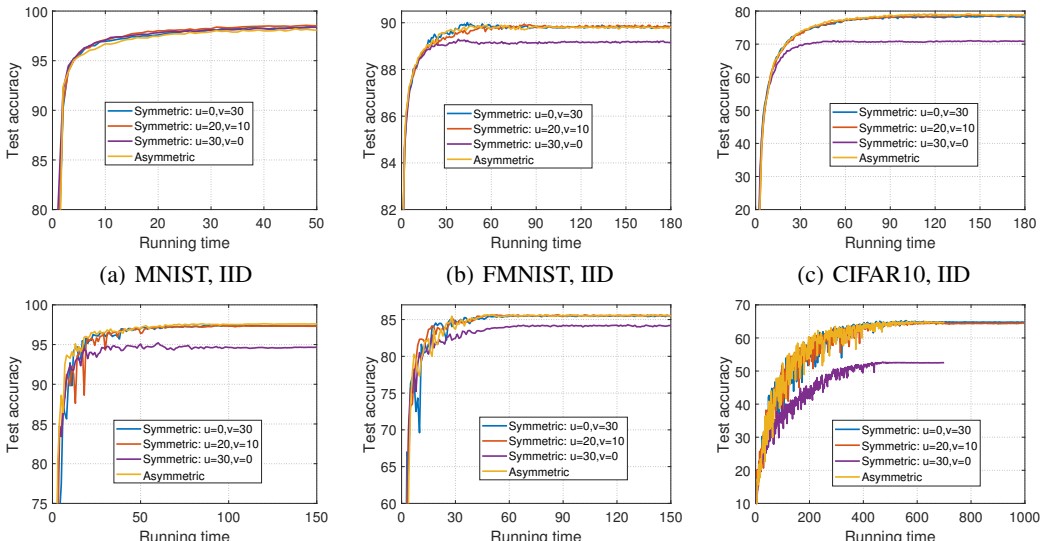

(a) MNIST, IID       (b) FMNIST, IID       (c) CIFAR10, IID

(d) MNIST, device nonIID, cell IID (e) FMNIST, device nonIID, cell IID (f) CIFAR10, device nonIID, cell IID

Figure E.1: Performance of FedMes in an asymmetric cell topology.

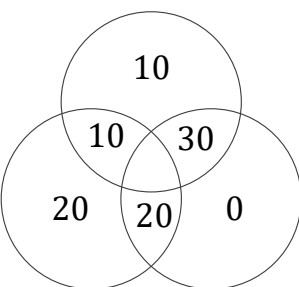

Figure E.2: Asymmetric topology with $L = 3$ cells.

# F  COOPERATION BETWEEN EDGE SERVERS

In the experiment section of the main manuscript, it is assumed that adjacent edge servers select the same set of devices from the overlapping area by cooperating with each other. In practical scenarios, we can imagine two different situations where this cooperative federated learning is available. First, if user mobility is not very high, the system proactively determines the candidate set of devices available to participate in the current federated learning process. Then, we can design a protocol that provides a rule of selecting devices in overlapping areas. For example, devices whose uplink SNRs with adjacent edge servers are larger than a given threshold value consist of the candidates set first. The candidate devices can be arranged in order of high average of uplink SNRs with adjacent edge servers, denoted by $d_1, \ldots, d_K$ where $d_i$ is the device whose average uplink SNR is the $i$-th highest, when $K$ devices are in the candidate set. Then, we can use the protocol that lets $d_1, \ldots, d_N$ participate in the first round of federated learning process. For the second round, $d_{N+1}, \ldots, d_{2N}$ can participate. Likewise,

a variety of protocols can be designed to determine which devices will participate in each round of federated learning process. If all edge servers agree to use certain protocol for selecting devices in the overlapping areas before the federated learning process begins, then we don't have to consider cooperation among adjacent edge servers in every round of learning process.

Second, even though most studies on federated learning (including this manuscript) assume that edge servers select devices unilaterally, participating for federated learning definitely sacrifices the devices' resources and battery. Therefore, in practice, edge servers have to request each device to participate first, and the devices will respond to it. In this process, we can find some devices that are willing to help cooperation among adjacent edge servers. This request-response process is required for our method as well as the conventional federated learning; therefore, it does not generate the additional latency for selecting the devices in overlapping areas. Thus, we can conclude that we don't have to consider the additional latency for selecting the same set of devices in the overlapping areas by cooperation among adjacent edge servers.

