# OpenReview forum: "FedMes: Speeding Up Federated Learning with Multiple Edge Servers"
_ICLR.cc/2021/Conference — Reject_

### Official Review · AnonReviewer4 · 2020-10-29
**Official Blind Review #4**

**Rating:** 4
**Confidence:** 4

**Review:**

This paper considers federated learning for edge devices with multiple wireless edge servers. The paper proposes FedMes to leverage devices in overlapping areas covered by multiple edge servers. In particular, in FedMes, if a device is in the coverage area of multiple edge servers, the device receives current models from all the edge servers covering it. Each device uses a (weighted) average of the models it receives as a starting point, and performs local updates (using SGD). A device broadcasts the updated model to multiple edge servers that cover the device. The key idea is that these devices in the overlapping coverage area act as ‘bridges’, and communication between edge servers is not required (until the final averaging step). The authors carry out experiments to evaluate FedMes, and compare against hierarchical federated learning of (Liu et al., 2019).

Strong points:

1. How the broadcast wireless nature can be leveraged in federated learning is a practically motivated question, and the proposed algorithm is quite simple.

2. The paper is well-written and easy to follow.

Major concerns:

1. The main premise of the paper is that communication with the central cloud server located at the higher tier of edge servers is costly and incurs significant delay. (One of the selling points of FedMes is that it does not require any backhaul traffic.) However, the authors do not provide much evidence for this premise. In fact, the bottleneck in wireless networks is typically the communication between edge devices and edge servers, and the backhaul communication is usually cheap and fast. It is not even clear why a central cloud server is required since edge servers can talk with each other periodically, and simply average the models. It will be important to give an evidence that communicating with a cloud server or between edge servers incurs large delay.

2. The paper only presents experimental evaluation, and the experiments make several simplifying assumptions. For instance, the latency with the cloud is modeled with simple one-round delay parameter t_c, and the one-round delay between a device and edge server is assumed to be 1. Further, the devices in overlapping coverage areas are assumed to be very symmetric. These assumptions do not seem to be practical, and it is difficult to assess the usefulness of FedMes over hierarchical FL. FedMes may perform better in these somewhat ideal conditions, but it would be important to consider practical aspects.

Specifically, (Liu et al., 2019) compute latency using a wireless model. Also, they consider various values of E and T_cloud. It will be useful to consider various values in the experiments for the fairness of comparisons. (Abad et al., 2020) also analyze end-to-end latency by considering the properties of the wireless nature. Overall, it will be important to consider similar models for latency other than taking a simplistic approach of one-round delay t_c.

Considering the above points, the contribution and novelty seem to be fairly limited. It will be helpful if authors can consider a more practical latency model, and/or provide more evidence for the values and assumptions used in experimental evaluations.

Other suggestions:

1. The authors cite latency sensitive applications, e.g., smart cars, to motivate faster training time. It will be helpful to clearly distinguish between training time and inference time. For instance, the 100 milliseconds latency considered in (Mao et al., 2017) is for training or inference?

2. In Fig. 2 and 3, what are the units of $t_c$ and the running time? When $t_c$ (delay between cloud and device) is increased, clearly FL and hierarchical FL will be slower. It will be important to give evidence for how these values are chosen. Also, what about the case when ES can talk with each other over backhaul; does it incur similar delay as talking to a cloud server?

3. In experiments, it is assumed that adjacent edge servers select the same set of devices from the overlapping area by cooperating with each other. It will be helpful to comment on the impact of this cooperation on latency as compared to sharing the (possibly compressed) models between edge servers.

4. Considering the same devices in the coverage area of an edge server throughout the training process does not seem to be practical. Would it be possible to extend the experiments to consider the case when the devices in the coverage area of an edge server follow some practically motivated distribution?

============= Post-Rebuttal Comments ===============
Thanks to authors for their response and efforts in updating the manuscript. Some of my concerns were addressed. However, I still think that the novelty is fairly limited. For example, additional experiments in Fig. 5 produce quite intuitive results in the sense that any scheme that yields smaller t_c will have higher accuracy at a specific time slot. FedMes reduces t_c with the assumption that it is faster to communicate between edge servers. Systems-level experiments to thoroughly study the effect on t_c will greatly improve the contributions.

---

> ### Author Response · Authors · 2020-11-17
> **Response to Reviewer 4 (Part 1)**
>
> $\textbf{Evidence/reference for the problem setup:}$ As described in (Mao et al., 2017), the supportable latency (for inference) of cloud-based systems is significantly larger compared to that of edge-based systems, due to the large communication delay between the cloud server and the devices. This large delay between the cloud and the devices directly affects the training time of cloud-based federated learning systems; as an example, in the original work on hierarchical FL (Liu et al., 2019), the communication latency to the cloud is assumed to be 10 times larger than that to the edge. The proposed FedMes which only utilizes communications between the edge and the devices, can significantly reduce the training time compared to cloud-based FL or hierarchical FL schemes that require costly communication between the cloud server and the clients. We tried to make this point clearer in the introduction of the revised manuscript.
>
> The central cloud server was necessary in previous works to obtain the full averaged model. If the edge servers communicate with each other in a decentralized manner via backhaul, the communication cost and delay increases with order of O(L^2), where L is the number of edge servers. If there exists a central cloud server, the order becomes O(L). Although there exist some algorithms that can reduce the communication cost in a decentralized setup, communicating with all other edge servers still incurs significant delay and complexity (compared to the scenario with a single cloud server for aggregation) in practice especially when the number of edge servers is large. This is one reason why the authors of (Liu et al., 2019) utilized a single cloud server for aggregation.
>
> $\textbf{Latency modeling:}$ We would like to first note that whatever computation/communication delay model we use, one can define the ratio of (one-round delay between the devices and the cloud) to (one-round delay between the device and the edge) as $t_c$, based on their system parameters. Now depending on the computation/communication delay model and parameters in the current system, the service provider can simply compute $t_c$ and decide which scheme to use (FedMes or Hierarchical FL), by observing the plots with different $t_c$ values in our manuscript. To sum up, whatever delay model we use, the latency can be parameterized as a single parameter $t_c$ based on the current system parameters, and this is the main reason why we utilized this modeling which captures all scenarios. We tried to make this point clearer in the revised manuscript.
>
> $\textbf{Symmetric overlapping coverage:}$ Following the reviewer’s comment, we performed additional experiments in an asymmetric cell topology. Now the number of devices are different across the overlapping region. Please refer to our new results in supplementary material. The results are consistent with the plots in the main manuscript, confirming the advantage of FedMes in practical asymmetric cell topology.
>
> $\textbf{Training/inference:}$ The supportable latency described in (Mao et al., 2017) is for inference. According to (Mao et al., 2017), the supportable latency of cloud-based systems is significantly large compared to the edge-based systems, due to the large communication delay between the cloud server and the devices. This large delay between the cloud and the devices directly affects the training time of cloud-based federated learning systems. Hence, FedMes which only utilizes the communication between the edge and the devices, can significantly reduce the training time compared to cloud-based systems; our FedMes can especially benefit latency sensitive applications, e.g., when a number of cars/drones should quickly adapt to the current situation by cooperation (federated learning) and make the right decision. We tried to make this point clearer in the introduction of the revised manuscript.
>
> $\textbf{Comments regarding $t_c$:}$ One-round delay between the device and the edge is normalized as 1, which is the basic unit. $t_c$ is obtained based on this basic unit. As we described above, all kinds of delay models can be parameterized with this $t_c$. We adopted $t_c=5$ and $t_c=10$ based on the description on the previous study on supportable latency (Mao et al., 2017), and the communication time ratio that the authors used in (Liu et al., 2019). Moreover, in Fig. 5 of the revised manuscript we plotted the accuracies as a function of $t_c$, to observe the gain of FedMes with varying $t_c$. Please refer to our revised manuscript. Regarding the scenario where ESs communicate with each other over backhaul, please refer to our answer above.

---

> > ### Author Response · Authors · 2020-11-17
> > **Response to Reviewer 4 (Part 2)**
> >
> > $\textbf{Cooperation between edge servers:}$ In practical scenarios, we can imagine two different situations where cooperative federated learning is available. First, the system can proactively determine the candidate set of devices available to participate in the current federated learning process. Then, we can design a protocol that provides a rule of selecting devices in overlapping areas. If all edge servers agree to use certain protocol for selecting devices in the overlapping areas before the federated learning process begins, then we don’t have to consider cooperation among adjacent edge servers in every round of learning process. Second, even though most studies on federated learning (including this manuscript) assume that edge servers select devices unilaterally, participating for federated learning definitely sacrifices the devices’ resources and battery. Therefore, in practice, edge servers have to request each device to participate first, and the devices will respond to it. In this process, we can find some devices that are willing to help cooperation among adjacent edge servers. This request-response process is required for our method as well as the conventional federated learning; therefore, it does not generate the additional latency for selecting the devices in overlapping areas. Thus, we can conclude that we don’t have to consider the additional latency for selecting the same set of devices in the overlapping areas by cooperation among adjacent edge servers. We added the descriptions about this issue in supplementary material.
> >
> > $\textbf{Considering the same devices in the coverage area of an edge server:}$ The participating devices are not fixed for each coverage. At each global round, each edge server randomly selects a set of devices to participate. For the case where devices in each covered area change in time (e.g., due to mobility), we performed additional results with varying $v$, and asymmetric user distribution scenario. The overall results show that FedMes has significant advantage compared to others even in these various user distribution setups. We thank the reviewer for the suggestions.

---

### Official Review · AnonReviewer2 · 2020-10-29
**Interesting problem but lack of necessary motivation and novelty.**

**Rating:** 5
**Confidence:** 4

**Review:**

Summary:
This paper proposed to decentralize FL by using multiple edge servers (ES), each only covering a subset of user devices, and utilize devices in the overlapped region of ESs to assist model aggregation, which saves the need to communicate with higher tier ESs. This decentralized architecture for FL hopefully can be more communication-efficient than a cloud-based or a hierarchical FL framework, with comparable model performance.

This paper has its merits in leveraging devices in overlapped ES cells to assist model aggregation, which could be a practical FL setting in the near future given the advance of 5G techniques. Experimental results on several benchmarks revealed the efficacy of this approach compared with a hierarchical FL baseline. However, the nonnegligible pitfalls of this paper are 1) the novelty is limited, 2) the challenge that they aim to address is unclear, and 3) their algorithm objective is not well formulated. Moreover, as a purely empirical study, its experiment designs are quite simple, missing comparisons with more state-of-the-art baselines, which further weakens their contribution.

Pros:
+ The idea of leveraging overlapped ESs to decentralize FL is legitimate.
+ This paper shows promising experimental results compared with Hierarchical FL which requires more communication rounds.

Cons:
- Although following an interesting direction, the novelty of their proposed algorithm is limited, and the challenge of this setting is not clearly described. I feel it is a straightforward extension of FedAvg and Hierarchical FL to a decentralized scenario.
- Performance gain over the prior art (*hierarchical* FL) is marginal when multiple clouds are adopted ($T_{cloud}=5$).
- Important baselines, such as cloud-based FL (FedAvg), and local training (without FL), are missing.
  - 1) The main focus of this paper is to decentralize FL to improve communication efficiency, but the tradeoff between performance and communication efficiency is not shown in the paper.
  - 2) Since the task of binary classification is quite simple, it is likely that each local device can perform well without using information from other devices, so I consider it necessary to comparing with local training.
- The argument that FedMes is more communication-efficient than cloud-based FL is doubtful. FedMes actually requires more communication rounds, since critical users in the overlapped region need to communicate with multiple servers. Another argument in the paper that the communication can be done by broadcasting applies to cloud-based FL as well. A theoretical or empirical analysis of communication efficiency is needed.
- Unclear objective: the proposed algorithm is designed to optimize Eq(1). However, since there is no higher-tier central server to aggregate the models from ESs, it is doubtful whether Algorithm 1 truly aligns with the optimization of Eq(1). A different objective, such as those adopted by personalized FL, where multiple sets of weights are learned rather than a single set, seems to fit this scenario more than the canonical FL objective.
- The weight design in Eq(9) is more like engineering efforts than theoretically motivated.
- Robustness of the proposed algorithm: Although I agree with the argument that "even users in the non-overlapped region can help training in other ESs". However, their contributions heavily depend on the participants of users in overlapped regions, and their hyper-parameter choice in Eq(9). An unresolved question is, when the number of users in overlapped regions is quite small, does their effects on other ESs reduce greatly as well? As shown in Figure 4, when there is no overlap, the performance is worse than all other baselines. The current experiment setting of using ratio 1:2 for overlapped versus non-overlapped users can be too enthusiastic. What if only 10% of users are in the overlapped regions? A sensitivity analysis of this ratio would be very interesting to see.

Minor:
- The definition of  $t_c$(as the one-round delay) should be more clearly elaborated.
- Eq(6) - Eq(7): weighted average requires each user in the overlapped region knows the total number of training samples for its corresponding ESs, is this assumption practical? In cloud-based FL, only the central cloud knows the number of training samples for each device.
- Eq(4): it would be better to use a different notation instead of $k$ for the second equation on the RHS to avoid confusion.

---

> ### Author Response · Authors · 2020-11-17
> **Response to Reviewer 2 (Part 1)**
>
> $\textbf{Novelty and challenge:}$ We note that only a few prior works considered a FL system with multiple edge servers. However, all the schemes in these previous works require costly communication with the central cloud server which significantly slows down FL due to large delay between the cloud and the clients. To overcome this challenge, we proposed FedMes, which is the first work that considers multiple edge servers in a FL setup without requiring costly communication with the cloud server. We tried to make this point clearer in the introduction of the revised manuscript.
>
> $\textbf{Straightforward extension:}$ We stress that our scheme is not a straightforward extension of previous works, since the setup and the algorithms are totally different. First of all, our system model is not hierarchical since it does not utilize the cloud, and thus the algorithm works totally in a different way compared with the hierarchical FL scheme. The utilization of devices in the overlapping areas is especially unique to our work, which significantly reduces the running time of FL and saves the backhaul costs in practical systems.
>
> $\textbf{Performance gain over hierarchical FL with $T_{cloud}=5$:}$ We would like to point out that Tcloud is the number of aggregations at each server before the cloud server aggregates the models of all edge servers, not the number of clouds. Hence, hierarchical FL with Tcloud =1 corresponds to the cloud-based FL system. The number of cloud servers is one for all comparison schemes, while our scheme does not require the cloud server. The performance gain of FedMes over hierarchical FL depends on the parameter $t_c$. To evaluate the gain more clearly, we plotted accuracy of each scheme as a function of $t_c$. It can be seen that FedMes performs the best even with a small $t_c$ value, although the gain is smaller with smaller $t_c$. However, in practical regimes where the supportable latency of cloud-based systems are much larger than that of edge-based systems (as described in (Mao et al., 2017)), FedMes can provide significant advantage compared to other baselines. We added this discussion in Section 4 of the revised manuscript.
>
> $\textbf{Baselines:}$ As described above, hierarchical FL with Tcloud=1 corresponds to the cloud-based FL system, which we have already considered as a baseline. We did not consider the local training scheme since it has significantly lower performance and the comparison is not that meaningful. Even though each client has only two classes in a non-IID scenario, they should also classify other classes well that are not in their local. As an example, for applications like disease diagnosis in hospitals, each client should classify reliably for classes that are not just in its local site but also with other clients. The goal of federated learning is to tackle this problem. This is why we plot the test accuracies with the test data having all 10 classes, which would significantly degrade the performance of local training. Many previous works show that FedAvg outperforms local training especially in this non-IID scenario.
>
> $\textbf{Communication efficiency:}$ To be precise, our main claim and the motivation of FedMes was to improve latency, not communication efficiency. As stated in the introduction, inherent limitation of this cloud-based system is the long distance between the device and the cloud server, which causes significant propagation delay during model downloading/uploading stages. This is why we consider edge servers instead of cloud server. Since the number of devices within the coverage of an edge server can be limited, we consider a practical setup with multiple edge servers. To sum up, we propose FedMes tailored to this decentralized setup to 1) improve latency with only edge servers (without the cloud server) and 2) cover sufficiently large area with multiple edge servers having limited coverage. These statements can be found in the Introduction of our manuscript.
>
> $\textbf{Comparison with local training:}$ Please refer to our response above regarding the baselines.
>
> $\textbf{Communication efficiency and broadcasting:}$ As mentioned in our response above, the main argument is not about communication efficiency but about latency. Since the one-round delay of FedMes is smaller than that of cloud-based system, it has performance improvements by observing more global rounds. Also, broadcasting the aggregated model from the server to clients is obviously the same in both FedMes and the cloud-based FL. However, please note that our key point is not broadcasting from the server to clients, but broadcasting from the devices in the overlapping areas to adjacent edge servers. Therefore, even though devices in overlapping areas send their results to multiple edge servers, it does not require additional communication rounds in FedMes. In cloud-based systems, all devices send their models to a single cloud server so this kind of broadcasting does not happen.

---

> > ### Author Response · Authors · 2020-11-17
> > **Response to Reviewer 2 (Part 2)**
> >
> > $\textbf{Objective:}$ It is true that multiple sets of models are learned from multiple edge servers, but in the final step of the FedMes algorithm, the models of all edge servers are aggregated to obtain a final model, as in Eq(10). This process is required to obtain a final global model that can handle all classes in the system, which coincides with the goal in Eq(1). We show via experiments that FedMes achieves better performance than hierarchical FL and cloud-based FL which are designed to optimize Eq(1).
> >
> > $\textbf{Eq(9):}$ Yes, we proposed Eq(9) based on the insights, as we described in the paper, which has been confirmed by experiments.
> >
> > $\textbf{Smaller number of devices in the overlapped region:}$ We thank the reviewer for the valuable comment regarding the robustness of FedMes. In the original manuscript we considered three cases: (u=0, v=30), (u=20, v=10), (u=30, v=0). Two extreme points and one middle point. We added experiments with a smaller portion of overlapping devices than (u=20, v=10), i.e., the case with (u=25, v=5). Although there are some slight performance degradation, FedMes performs well even with a small portion of devices in the overlapping regions (i.e., small $v$). Especially in 5G and beyond systems where a large number of devices are in the overlapping regions, FedMes can provide significant advantage compared to other schemes. We added this discussion in Section 4 of the revised manuscript.
> >
> > $\textbf{Definition of $t_c$:}$ We added the description on $t_c$. This one-round delay include both the communication time for one-round trip between the server and the device, and the computation time at each device.
> >
> > $\textbf{Eq(6) - Eq(7):}$ Note that each ES knows the number of training samples from the received devices, as the cloud server does in cloud-based systems. Hence, each ES can simply send the number of data samples along with the aggregated models to the devices. Then, the devices can compute the weighted average as in Eq(6).
> >
> > $\textbf{Eq(4):}$ We used different notation instead of k in the revised manuscript.

---

> > > ### Comment · AnonReviewer2 · 2020-11-23
> > > **Feedback**
> > >
> > > I would like to thanks the authors for providing a response with more experimental results. It alleviates my concerns to some extent. However, questions regarding the motivation, novelty, and theoretical analysis of this paper are still unresolved in its current version.
> > > ​
> > > - Motivation not well-evidenced: The argument in the response that "our system model is not hierarchical since it does not utilize the cloud" (mentioned in the revised introduction as well) contradicts the fact that a final model by averaging is still performed (as in Eq 10). If a final model is required, then the difference between this work and previous Hierarchical FL is blurry. Actually, it makes more sense if a final global model is not needed, which aligns more with the claim of this paper.
> > >
> > > - Theoretical analysis regarding its convergence bound may help to extend the contribution scope of this work and make their argument more convincing.

---

> > > > ### Author Response · Authors · 2020-11-24
> > > > **Response**
> > > >
> > > > We would like to point out that our scheme constructs the final model when training is finished (according to Eq 10), which means that only one aggregation process is required in the final step. Compared to our scheme, hierarchical FL requires periodic aggregation at the cloud server during the whole training session. We thank the reviewer for the feedback.

---

### Official Review · AnonReviewer1 · 2020-11-06
**Review 1**

**Rating:** 5
**Confidence:** 4

**Review:**

## Summary
In previous federated learning literature, people usually assume there is only one cloud server communicating with all edge nodes/clients. However, since each server has its own coverage in practice, the latency between the server and clients out of the coverage can be pretty long. This paper focuses on reducing the communication cost in this practical setting. The authors propose to use multiple edge servers, which have overlapped coverages. The clients in the overlapping areas will receive model parameters from multiple server and return the average model back. This can help to mix the information between different edge servers. Experiments on MNIST, EMNIST, and CIFAR10 datasets validate the effectiveness of the proposed algorithm: FedMes.

This paper provides novel insights into a practical setting of FL. But its experimental setting is too simple and there's no theoretical guarantees. These make the evaluations to be less convincing.

## Pros
- The multi-server setting is interesting and practical. It hasn't been well-studied before. This paper makes the initial step along this direction.

## Cons
1. The simulation plots only show accuracy-versus-time. It would be great to show accuracy-versus-round as well. In figure (3), the purple line doesn't converge yet due to its high running time per round. It is unclear whether the purple line can achieve a higher final accuracy than other methods. Intuitively, there should be a trade-off: although communicating with a single cloud server costs a lot, it can ensure all local models to be the same and may have better final accuracy. This trade-off isn't clearly discussed in the paper.
2. In practice, whether to use the proposed algorithm should depend on the latency per round as well as the topology of the overall network. For example, in the cases where the latency $t_c$ is very small or the number of overlapping clients $v$ is very few, then the performance of FedMes may not beat Hierarichical FL (T_cloud=1). In the paper, it's unclear how the performance of FedMes changes along with $t_c$ and the topology of the network, as the authors only consider 2 values for $t_c$ and 3 values for $v$. It would be better to have some plots showing how the accuracy changes with these parameters. It would help people to have a better idea on when to use this algorithm.
3. This paper doesn't provide any theoretical guarantee for the proposed algorithm. It would be nice to see how the error bound scales with $t_c, u, v$. The analysis techniques for FedAvg and decentralized averaging is very mature. The authors can take a look at [1-4] to see whether FedMes can be easily analyzed by these frameworks.
4. In the experiments, the authors only consider a symmetric topology among clients with only 3 edge servers. One can also generate some random topologies with arbitrary overlapping clients to test the performance of the algorithm. This kind of evaluation would be more convincing. Since there is no theoretical guarantees, only empirical results on a simple symmetric topology is not enough. It's possible that the proposed algorithm only works for this special case.

## Post-rebuttal comments
Thanks the authors for the clarifications! Most of my concerns are addressed. But the newly added asymmetric topology is still very benign and there is only 3 cells. I agree with other reviewers that this paper can be further improved.

## References:
[1] Stich. Local SGD Converges Fast and Communicates Little. ICLR 2019
[2] Li et al. On the convergence of FedAvg on non-iid data. ICLR 2020
[3] Wang and Joshi. Cooperative SGD: a unified analysis for the design and analysis of communication-efficient SGD Algorithms. ICML 2019 workshop
[4] Khaled et al. Tighter Theory for Local SGD on Identical and Heterogeneous Data. AISTATS 2020

---

> ### Author Response · Authors · 2020-11-17
> **Response to Reviewer 1**
>
> $\textbf{Accuracy-versus-round plot:}$ We thank the reviewer for this valuable comment. We added the accuracy-versus-round plots in revised manuscript for all datasets and all distribution setups. It can be seen that FedMes with both (u=0, v=30) and (u=20, v=10) achieve the same final accuracy with the cloud-based system (purple line), except for only one case. For CIFAR-10 in a (device non-IID, cell non-IID) setup, FedMes with (u=0, v=30) achieves the same final accuracy with the purple line but the case with (u=20, v=10) does not. This indicates that a larger number of devices should be in the overlapping areas to act as bridges, especially when the data distributions across the cells are non-IID and the dataset is relatively complex. Thanks to the reviewer’s comment, we added this results and discussions in the supplementary material of our revised manuscript.
>
> $\textbf{Performance with varying $t_c$:}$ Based on the reviewer’s comment, we plotted accuracy as a function of $t_c$. It can be seen that FedMes performs the best even with a small $t_c$ value. Especially in practical regimes where the supportable latency of cloud-based systems are much larger than that of edge-based systems (as described in (Mao et al., 2017)), FedMes can provide significant advantage compared to other baselines. Thanks to the reviewer’s comment, we added this discussion in Section 4 of the revised manuscript.
>
> $\textbf{Performance with varying $v$:}$ Regarding the number of clients $v$, we considered three cases in the original manuscript: (u=0, v=30), (u=20, v=10), (u=30, v=0). Two extreme points and one middle point. We added experiments with a smaller portion of overlapping devices, i.e., the case with (u=25, v=5). Although there are some slight performance degradation, FedMes performs well even with a small portion of devices in the overlapping regions (i.e., small $v$). Especially in 5G and beyond systems where a large number of devices are in the overlapping regions, FedMes can provide significant advantage compared to other schemes. We added these discussions in Section 4 of our revised manuscript.
>
> $\textbf{Theoretical guarantee:}$ We appreciate this valuable comment. We have struggled to show the convergence of our proposed FedMes algorithm mathematically, but unfortunately, it appears to be tricky because the initial models of edge servers are different from each other every round except for the first round. Instead, as the reviewer commented above and below (comments 1, 2, 4), we tried to provide as many simulation results as possible in a variety of scenarios (varying $t_c$, varying $v$ and an asymmetric topology). As the original manuscript shows, the proposed FedMes can catch up the accuracy of the hierarchical scheme which is the state-of-the-art while saving backhaul costs between a central server and edge servers and reducing the overall training time. Therefore, we believe that the FedMes would be valuable in various practical scenarios even though its convergence bound is not theoretically derived.
>
> $\textbf{Asymmetric topology:}$ Thanks to the reviewer’s comment, we performed additional experiments in an asymmetric cell topology. Now the number of devices are different across the overlapping region. Please refer to our new results in supplementary material. The results are consistent with the plots in the main manuscript, confirming the advantage of FedMes in practical asymmetric cell topology.

---

### Author Response · Authors · 2020-11-17
**General comments to reviewers**

First of all, we would like to thank the reviewers for their efforts and constructive suggestions, which have greatly helped us to improve the paper. We have worked them into the revised version of the paper. The main changes are as follows:

1. We tried to make the motivation and problem formulation be clearer.
2. We performed additional experiments to validate our scheme further (asymmetric cell topology, varying $v$, $t_c$).

All the changes in the revised paper are marked in blue color. For more details, please refer to our official comments corresponding to each review.

---

### Decision · Program_Chairs · 2021-01-07
**Final Decision**

**Decision:**

Reject

**Comment:**

The paper studies the benefit of having multiple servers (with partial coverage) in increase the training speed and latency in Federated Learning.  Of course optimization/learning in the multi-server setting comes with a number of challenges which the authors seek to address via novel algorithmic procedures (e.g. FedMes).  I believe the paper is suggesting an important, and potentially impactful, methodology to improve the training speedup/latency of FL. I also acknowledge the additional experiments provided by the reviewers which were quite helpful in addressing some of the concerns. However, as the paper mainly relies on experimental studies to evaluate the performance of the proposed methods, the reviewers (and myself) believe that the paper needs some more investigation in which (i) some of the assumptions (e.g. faster communication between the servers) are either removed or validated; and (ii) more complicated  topologies are considered.